# Knowledge and Awareness of HPV, the HPV Vaccine and Cancer-Related HPV Types among Indigenous Australians

**DOI:** 10.3390/ijerph21030307

**Published:** 2024-03-06

**Authors:** Lucy Lockwood, Xiangqun Ju, Sneha Sethi, Joanne Hedges, Lisa Jamieson

**Affiliations:** Australian Research Centre for Population Oral Health (ARCPOH), Adelaide Dental School, University of Adelaide, Adelaide 5005, Australia; lucy.lockwood@student.adelaide.edu.au (L.L.); sneha.sethi@adelaide.edu.au (S.S.); lisa.jamieson@adelaide.edu.au (L.J.)

**Keywords:** human papilloma virus, Indigenous Australian, knowledge and awareness, HPV vaccine

## Abstract

Human Papilloma Virus (HPV) infection is a common, preventable, sexually transmitted disease with oncogenic potential and increasing incidence. This study aimed to gain an understanding of the knowledge and awareness of HPV, the HPV vaccine, and HPV-related cancers, and to evaluate the relationship between participant factors and HPV knowledge, vaccination uptake, and high-risk HPV (16/18) infection, among Indigenous Australians. Data from the 12-month follow-up of a longitudinal cohort study were utilized, involving 763 Indigenous Australian adults in South Australia. The data analysis found that the mean 7-item HPV knowledge tool (HPV-KT) score was 2.3 (95% CI: 2.1–2.4), HPV vaccination prevalence was 27.0% (95% CI: 23.6–30.5) and oral HPV 16/18 infection was 4.7% (95% CI: 3.2–6.2). Multivariable log-Poisson regression models showed ratios of approximately 1.5 times higher HPV-KT scores in females, previous recreational drug users, those who had self-rated as having excellent, very good or good general health and who had heard of HPV; and participants who were not HPV vaccinated had 0.8 times (MR = 0.8, 95% CI: 0.7–0.9) lower HPV-KT scores than their counterparts. The findings suggest that culturally safe education strategies are a necessary investment to improve vaccination coverage among Indigenous Australians and to reduce the impact of HPV and related cancers.

## 1. Introduction

Human Papilloma Virus (HPV) infection is a common, preventable, sexually transmitted disease with an increasing incidence globally [1]. With an estimated incidence of 80% among women and men by 45-years-old, nearly all sexually active adults contract HPV during their lifetime, with the majority successfully clearing without clinical presentation [2,3]. The small, double-stranded DNA virus infects basal cells in the epithelium of the skin and mucous membranes, possibly leading to presentations such as warts and cancers of the cervix, oropharynx, anus, penis and vulva [4,5,6,7]. HPV can be transferred during sexual intercourse, oral sex, kissing, vaginal birth, autoinoculation and from fomites, where a person has been exposed to contaminated items for a prolonged period [4,8]. The number of sexual partners, age of sexual debut and sexual behaviors change the HPV infection risk [8,9]. Over 200 subtypes of HPV have been detected and categorized into high-risk (hr) and low-risk (lr) types [8]. HPV types 16 and 18 are within the high-risk group and most strongly associated with cancer, being detected in the majority of HPV-positive oral and oropharyngeal cancers (OPCs) and 70–80% of cervical cancers globally [4,5,6]. Previously, the most common HPV-driven cancers were cervical cancers, but these have been surpassed by OPCs, which involve the tonsillar area, base of tongue and soft palate [10,11,12]. The survival rates associated with HPV-positive OPCs appear to be better than non-HPV-related OPCs [6].

The HPV vaccine is effective against the two most common hr HPV types when administered prior to exposure [13]. Australia was the first country to fund a National HPV Vaccination Program, with delivery to girls starting in 2007 and boys in 2013. The school-based program delivers the vaccine to Year 7 adolescents (aged 12–13 years). Those who missed out can receive the vaccine for free up to 26 years old [14]. Initially, the vaccine program was a 3-dose quadrivalent Gardasil HPV vaccine, which was replaced with a 2-dose nonvalent HPV vaccine from 2018. Since February 2023, the program has adopted a single-dose system using a Gardasil^®^9 vaccination (effective against HPV types 6, 11, 16, 18, 31, 33, 45, 52 and 58) [15,16]. The national vaccination coverage for Indigenous Australian adolescents has been high, with 2016 data showing over 80% of girls and boys received at least 1 dose; however, course completion was low [15]. With the transition to single-dose administration, coverage is likely to improve.

Aboriginal and Torres Strait Islander people (hereby respectfully referred to as Indigenous Australians) have resided in the country for over 50,000 years and represent 3.3% of Australia’s population [5]. The connection to land, family, culture and spirit often contributes to overall well-being [5]. Many Indigenous cultures educate younger generations about history, country and health through storytelling. However, in the past, Indigenous groups were discouraged from discussing sexual health, making it difficult to share important information with children and grandchildren [17]. Globally, Indigenous people experience disproportionate health inequalities compared to non-Indigenous people, as well as the ongoing effects of colonization and systemic racism [18,19]. Despite efforts to close the health equity gap in Australia, Indigenous Australians experience higher incidence rates of cancer and have less chance of surviving five years after diagnosis compared to non-Indigenous Australians [20].

Generally, screening and vaccination have reduced the incidence of HPV infection; however, evidence shows that Indigenous Australians are still experiencing higher HPV infection rates [1,5]. In 2020, the Australian Research Centre for Public Oral Health (ARCPOH) reported that 35.5% of Indigenous Australian participants were positive for any oral HPV infection [5]. The prevalence was found to be 15 times that among young non-Indigenous Australians and 4.7 times that reported by Antonsson et al. in a systematic review of 9 international papers [5]. Multiple barriers to receiving the HPV vaccine among Indigenous groups have been found, impacting administration at its most effective time. These barriers include mistrust of health care systems, lacking understanding of HPV, resource constraints, service infrastructure gaps, staff shortages, sensitivity regarding sexual health promotion and concerns regarding medical practitioner abilities [17,21,22]. There is limited evidence regarding HPV-related OPCs or HPV vaccine uptake among Indigenous Australians [9].

It is well understood that lack of information contributes to vaccine reluctance, with many studies suggesting that increasing awareness of HPV, its vaccine and its association with cancer can contribute to better vaccine uptake and infection prevention [18,21,23,24,25,26,27]. A study involving American Indian adolescents found that HPV vaccine initiation was higher among those whose parents were aware of the vaccine and had received a recommendation from their medical practitioner for their child to receive it [28]. Researchers have suggested that Indigenous participants have “little” knowledge about HPV, its mode of transmission, the HPV vaccine or its oncogenic potential [28]. However, there was no evidence quantifying knowledge of HPV, its vaccine and related cancers among Indigenous Australians, or the individual factors associated with poorer knowledge.

This study aimed to gain an understanding of the knowledge and awareness of HPV, the HPV vaccine, and HPV-related cancers, and to evaluate the relationship between participant characteristics and knowledge of HPV, HPV vaccination uptake and cancer-related high-risk HPV (HPV16/18) infection, among Indigenous Australian adults. We hypothesized that Indigenous Australians have limited knowledge and awareness regarding HPV, the HPV vaccine, and HPV-related cancers, and that multiple factors contribute to this.

## 2. Materials and Methods

### 2.1. Study Design, Sample Size and Data Collection

The Australian Research Centre for Public Oral Health (ARCPOH) embarked on a large longitudinal cohort study in 2018 (hereby referred to as the ARCPOH HPV study), with the aim of yielding population estimates of oral oncogenic HPV and the proportion of OPCs caused by HPV and understanding the impact of HPV vaccination on incidence rates among Indigenous Australians [9]. Baseline data collection was conducted between February 2018 and January 2019, involving a large convenience sample (n = 1011) of Indigenous Australian adults in South Australia aged 18+ years who were recruited through Aboriginal Community Controlled Health Organisations (ACCHOs). This study had the oversight of the ACCHOs and an Indigenous Reference Group (IRG), who advised on all aspects of the project, staff employment, participant recruitment, data collection, analysis, and feedback. This cross-sectional sub-study used data from the ARCPOH HPV study baseline and 12-month follow-up.

Our findings are reported according to the Strengthening the Reporting of Observational Studies in Epidemiology (STROBE) guidelines.

### 2.2. Variables

#### 2.2.1. Outcome Variable

##### Primary Outcome

The primary outcome was the HPV knowledge tool (HPV-KT) scores. Waller and Ostini developed and validated a 29-item HPV knowledge tool in 2011 that measured “General HPV knowledge”, “HPV testing knowledge” and “vaccination knowledge” among Australian, American, and British adults [29,30]. We developed, in consultation with the IRG, an adaptation of the knowledge tool to a shorter 10-item version (see Appendix A) [30]. Statistical analyses of the psychometric properties of the 10-item HPV-KT among Indigenous Australians found overlap and low stability for several questions. Exclusion of these questions (questions 5, 6 and 7) resulted in a 7-item HPV-KT. The 7-item HPV-KT showed adequate psychometric properties and is recommended for use among Indigenous Australians with consideration of local factors and personal circumstances [30]. Therefore, the 7-item score (hereby referred to as the HPV-KT) was used as the primary outcome. This instrument contains 7 items: HPV (1) is very rare; (2) always has visible signs and symptoms; (3) can be passed on by genital skin-to-skin contact; (4) has many types; (8) most sexually active people will get HPV at some point in their lives; (9) can be carried without for many years; (10) can cause cancer in men. Each item had three options (1 = “True”, 2 = “False” and 3 = “Don’t know”). Each answer was objective. After re-coding the correct response = 1, and incorrect and “Don’t Know” responses = 0, the summary scores ranged from 0 to 7. A higher score indicated higher knowledge.

##### Secondary Outcomes

HPV vaccination

Participants were asked at baseline if they had received the HPV vaccination. The 12-month follow-up survey asked more specific questions about the HPV vaccine, regarding the number of doses and location of vaccine administration (see Appendix A). The “Had HPV vaccine” variable was derived by combining the baseline and 12-month follow-up data. “Yes” meant that participants had received one or more vaccine(s) from baseline to the 12-month follow-up.

Oral high-risk HPV types 16 and 18 infection

Saliva samples were collected from participants at baseline and 12-month follow-up for DNA analysis. Β-globin polymerase chain reaction (PCR) tests with the primers PCO3/4 were carried out on all the samples to ensure enough cells to detect human DNA were present, and that PCR inhibiting agents were absent. Saliva samples that were β-globin-positive were included in the data analysis. Β-globin is a test of DNA integrity; any samples with a negative β-globin were invalid. HPV analysis was performed using a nested PCR system MY09/11 and GP5þ/6þ to detect a large spectrum of mucosal HPV type, which included high-risk HPV types 16 and 18, which have oncogenic potential in mucosal tissue [1,5].

#### 2.2.2. Covariates

Covariate factors were provided by participant self-reported data, which were characterized into sociodemographic characteristics (age, gender, residential location, educational level, household income, health care card holder, car ownership), health and health-related behaviors (smoking status, alcohol consumption, non-tobacco smoking, recreational drug use, ever had tonsils out, self-rated general health, self-rated oral health), sexual behaviors (had passionately kissed, ever given oral sex, given oral sex age, number of oral sexes given, ever received oral sex, received oral sex age, number of oral sexes received, had sexual intercourse, had sexual intercourse age, number of sexual intercourses, current relationship status) and HPV knowledge and related behaviors (ever had HPV, had heard of HPV, self-rated HPV knowledge) (see Appendix A).

### 2.3. Statistical Analysis

The statistical analysis included descriptive analyses and multivariate analyses. The descriptive analysis began with the computation of univariate statistics describing the frequency and percentage covariates and the associated mean and 95% confidence interval (CI) of the HPV-KT scores, and the prevalence of received HPV vaccine and HPV 16 and/or 18 infections. Statistically significant differences were denoted by 95% CIs that did not overlap.

The multivariate analysis included multivariate logistic regression models with Poisson/binomial distribution estimation generation to calculate the risk indicators. Unadjusted and adjusted mean ratios (MRs) and their 95% CI were calculated for the mean HPV-KT scores. Blocks of covariates were entered into multivariable models in six steps: model 1 was the crude model, the sociodemographic factors characteristics were entered into model 2, health and health-related behaviors were entered into model 3, sexual behaviors were entered into model 4; and HPV vaccine, oral HPV 16 and 18 infection, HPV knowledge and related behaviors were put into model 5, with the final model 6 (full model) comprising all the factors.

The data files were managed, and summary variables were computed and analyzed using SPSS software version 19 and SAS software version 9.4 (SAS 9.4, SAS Institute Inc., Cary, NC, USA).

## 3. Results

Of the 763 participants at the 12-month follow-up, 747 completed the HPV-KT questions. Table 1 and Table 2 display the participant characteristic proportions and associations with the HPV-KT scores, HPV vaccination status, and oral HPV type 16 and/or 18 infections.

### 3.1. Participant Characteristics

A higher proportion of participants were in the older (>37) age group, females, resided in non-metropolitan areas, reported the highest level of education as high school or less, received Centrelink payments, were health care card holders, and owned their own car. A higher proportion of participants self-reported as current smokers, never having tonsils out and excellent, very good or good general and oral health. The majority of participants had passionately kissed ≥4 people, given oral sex, received oral sex, had sexual intercourse and were in a long-term relationship. A high proportion of participants had heard of HPV but self-rated their HPV knowledge as poor or fair (Table 1).

### 3.2. Mean HPV-KT

The mean HPV-KT score was 2.3. Participant groups with statistically higher HPV-KT scores were those who had an education level of trade or over (2.9), a job (2.9), were non-health care card holders (2.8), been diagnosed with HPV (4.3), self-rated general health as excellent, very good, or good (2.4), received oral sex (2.5), had heard of HPV (2.9) and those who had good or very good self-rated HPV knowledge (3.7) (Table 1).

Analysis of the HPV-KT answers showed that only one question (A9) had a correct response percentage above 50% (see Appendix A). The percentage of correct responses ranged from 22.4% to 51.3%.

### 3.3. HPV Vaccination Rate

The prevalence of participants having received the HPV vaccine was 27.0%. There was a statistically significantly higher prevalence of vaccination among females (33.1%) and the younger age group (52.7%). Participants who reported having heard of HPV and self-rated their HPV knowledge as poor or fair had higher vaccination rates compared to those who reported very poor or no knowledge of HPV (Table 2).

### 3.4. HPV 16 and/or 18 Infections

The prevalence of oral HPV 16 and/or 18 infection was 4.7%, with a statistically higher prevalence among participants living in metropolitan areas (8.1%) (Table 2).

### 3.5. HPV-KT Mean Ratios

Table 3 presents the association between the mean HPV-KT scores and risk indicators in the unadjusted, group adjusted and fully adjusted models.

Participants who had not received the HPV vaccine scored 0.7 times lower than vaccine recipients. Those who were <37 years old, female, held a trade or over, a job, were non-health care card holders or car owners scored between 1.2–1.6 times higher than their counterparts.

Current smokers scored lower than participants that never smoked. Participants who had previously used recreational drugs scored 1.2 times higher than those who had never used recreational drugs. Participants who have had their tonsils out, self-rated their general health as excellent, very good or good, had passionately kissed four or more people, given oral sex, or received oral sex scored between 1.2–1.5 times higher than their counterparts.

Participants who have had HPV, or heard of the infection, scored approximately two times higher than those who were unsure if they had been diagnosed with, or heard of HPV. Participants with good or very good self-rated HPV knowledge scored nearly three times higher than those who had never heard of or who had very poor self-rated HPV knowledge.

## 4. Discussion

The mean HPV-KT score was low (2.3), supporting the hypothesis that Indigenous Australians have limited knowledge and awareness regarding HPV, the HPV vaccine, and HPV-related cancers. Multiple contributing factors, such as education, health care card status, HPV infection, general health, sexual behavior and self-rated HPV knowledge, were identified.

All the HPV-KT questions, except question nine, had more “Don’t Know” responses than “True” and “False” combined. This highlights that many participants did not have incorrect knowledge but were rather unaware of HPV, its high occurrence, transmission, and oncogenic potential. The two lowest scoring questions were “HPV is very rare” and “HPV always has visible signs or symptoms”, indicating the need to improve education regarding the commonness of HPV and asymptomatic infection.

The total prevalence of HPV vaccination was 27%. Most national and international estimates focus on vaccination coverage among adolescents, making a comparison of the vaccination rate among Indigenous Australian adults difficult. The HPV vaccination rates were significantly higher among the younger age group, a result of the HPV vaccination program commencing in 2007. Until 2013, only females were provided with HPV vaccination in school, likely contributing to the difference in vaccination coverage among males and females. The vaccination rates were higher among those who had previously heard of and had some knowledge regarding HPV (35.5%), supporting the long-term idea that education facilitates vaccination.

The HPV 16 and/or 18 infection prevalence among the Indigenous Australian participants (4.7%) exceeded rates reported among young Australians (1.3%) and an international systematic review involving nine papers (average HPV 16 prevalence of 1.6%) [31,32].

Indigenous populations face multiple barriers when considering receiving the HPV vaccination. These barriers are a consistent trend, with Indigenous people in Canada, Aotearoa New Zealand, and the USA all reporting low awareness of the HPV vaccine and low access to culturally safe clinics [24]. When strategizing how to improve Indigenous population health, it is important to consider Indigenous health practices and methods of sharing information. International studies highlight that the development of culturally sensitive education strategies, based on the current reported barriers to and facilitators of vaccination, can encourage parents to make informed decisions about vaccinating their children [17,33]. Several strategies have been suggested, including collaboration with Indigenous communities to create educational programs that adopt a verbal approach, community-based vaccination programs, individual counselling, and increased health care provider knowledge and recommendations [17,22,33,34,35].

The strengths of this study include the engagement with Indigenous Australian communities through partnerships and involvement of the Indigenous Reference Group, a large sample size, and psychometric analysis of the 10-item HPV-KT prior to use. A study limitation was the cross-sectional design, preventing the testing of causal hypotheses. Psychometric analysis of the 7-item HPV-KT found poor reliability for the questions associated with “commonness of HPV”. Despite this, it was recommended for use among Indigenous Australians, with the use of a subscale listed as a limitation.

We hope that the findings of this project can be used by policy makers to allocate funding into culturally sensitive education strategies. A lack of knowledge has been found in our study; further research can be conducted to determine whether the lack of knowledge among Indigenous Australians is contributing to the higher HPV and HPV-related OPC incidence among this population.

## 5. Conclusions

The responses to the HPV-KT indicate generally poor understanding regarding HPV among Indigenous Australians, with variation among sociodemographic and health behavior groups. Participants who had heard of HPV and highly rated their general health demonstrated better HPV knowledge and were more likely to be vaccinated against HPV. The oral HPV 16 and 18 infection rates among Indigenous Australian Adults exceeded national and international estimates.

Improved HPV and vaccine education would benefit many populations; however, the findings of this sub-study and a larger prospective study suggest that culturally safe education strategies are a necessary investment to improve HPV vaccination coverage among Indigenous Australians and reduce the impact of HPV and related cancers.

## Figures and Tables

**Table 1 ijerph-21-00307-t001:** Participant demographic characteristics and associated mean HPV-KT scores among Indigenous Australian adults.

		Number	% (95% CI)	Mean (95% CI)
**Total**		763		2.29 (2.14–2.44)
**Had HPV vaccine**	No	456	**73.0 (69.5–76.5)**	2.08 (1.89–2.28)
	Yes	169	**27.0 (23.6–30.5)**	3.14 (2.83–3.45)
	Missing	138		
**Oral HPV 16/18 infection**	Yes	36	**4.72 (3.21–6.23)**	2.39 (1.69–3.09)
	No	727	**95.3 (93.8–96.8)**	2.29 (2.13–2.44)
	Missing	0		
**Sociodemographic characteristics**				
**Age group**	≥37	422	55.3 (51.7–58.9)	2.14 (1.94–2.35)
	<37	341	44.7 (41.1–48.3)	2.48 (2.25–2.70)
	Missing	0		16
**Gender**	Male	243	31.8 (28.6–35.3)	2.01 (1.76–2.27)
	Female	520	68.2 (64.7–71.4)	2.42 (2.23–2.61)
	Missing	0		16
**Location**	Non-metro	464	60.9 (57.3–64.4)	2.25 (2.06–2.44)
	Metro	298	39.1 (35.6–42.7)	2.36 (2.11–2.60)
	Missing	1		17
**Education level**	High school or less	499	66.4 (62.9–69.7)	**1.99 (1.81–2.16)**
	Trade or over	253	33.6 (30.3–37.1)	**2.90 (2.63–3.18)**
	Missing	11		27
**Income**	Centrelink	558	74.3 (71.0–77.4)	**2.06 (1.89–2.23)**
	Job	193	25.7 (22.6–29.0)	**2.91 (2.60–3.22)**
	Missing	12		28
**Health care card holder**	Yes	560	77.3 (74.1–80.3)	**2.17 (2.00–2.34)**
	No	164	22.7 (19.7–25.9)	**2.80 (2.45–3.16)**
	Missing	39		54
**Car ownership**	No	315	41.6 (38.0–45.2)	2.13 (1.90–2.35)
	Yes	443	58.4 (54.8–62.0)	2.40 (2.20–2.61)
	Missing	5		21
**Health-related behaviors**				
**Smoking status**	Current smoker	425	59.0 (55.3–62.6)	2.14 (1.96–2.34)
	Ex-smoker	97	13.5 (11.1–16.2)	2.78 (2.32–3.24)
	Never smoker	198	27.5 (24.3–30.9)	2.47 (2.19–2.76)
	Missing	43		58
**Consumes alcohol**	Daily	27	3.6 (2.4–5.3)	2.19 (1.34–3.03)
	Weekly	179	24.2 (21.1–27.4)	2.31 (2.00–2.61)
	Monthly	265	35.8 (32.3–39.3)	2.27 (2.01–2.54)
	Never	270	36.4 (33.0–40.0)	2.32 (2.07–2.57)
	Missing	22		38
**Non-tobacco smoking**	Currently	89	12.4 (10.1–15.0)	2.18 (1.76–2.60)
	Used	135	18.8 (16.0–21.8)	2.56 (2.19–2.94)
	Never	495	68.8 (65.3–72.2)	2.27 (2.09–2.46)
	Missing	44		59
**Recreational drug use**	Currently	151	20.1 (17.3–23.1)	2.36 (2.02–2.71)
	Used	254	33.8 (30.4–37.3)	2.51 (2.24–2.78)
	Never	347	46.1 (42.5–49.8)	2.08 (1.86–2.29)
	Missing	11		27
**Ever had tonsils out**	Yes	101	13.6 (11.2–16.3)	2.55 (2.11–2.99)
	No	598	80.7 (77.7–83.5)	2.28 (2.11–2.45)
	Unknown	42	5.7 (4.1–7.6)1	1.76 (1.12–2.41)
	Missing	22		38
**Self-rated general health**	Excellent/very good/good	577	77.2 (74.1–80.2)	**2.41 (2.23–2.58)**
	Fair/poor	170	22.8 (19.8–25.9)	**1.91 (1.61–2.21)**
	Missing	16		32
**Self-rated oral health**	Excellent/very good/good	480	65.0 (61.5–68.5)	2.29 (2.10–2.47)
	Fair/poor	258	35.0 (31.5–38.5)	2.30 (2.03–2.57)
	Missing	25		41
**Sexual health behaviors**				
**Had passionately kissed**	≥4	457	66.8 (63.1–70.3)	2.44 (2.24–2.64)
	<4	227	33.2 (29.7–36.9)	2.05 (1.79–2.32)
	Missing	79		94
**Ever given oral sex**	Yes	457	66.8 (63.1–70.3	2.46 (2.25–2.66)
	No	227	33.2 (29.7–36.9)	2.00 (1.75–2.26)
	Missing	79		94
**Given oral sex age**	<16	100	22.2 (18.5–26.4)	2.55 (2.09–3.02)
	≥16	350	77.8 (73.6–81.5)	2.42 (2.19–2.69)
	Missing	313		324
**Number of oral sexes given**	>3	197	43.5 (38.9–48.2)	2.62 (2.30–2.93)
	≤3	256	56.5 (51.8–61.1)	2.33 (2.06–2.60)
	Missing	310		321
**Ever received oral sex**	Yes	454	67.6 (63.9–71.1)	**2.45 (2.25–2.65)**
	No	218	32.4 (28.9–36.1)	**1.97 (1.71–2.23)**
	Missing	91		105
**Received oral sex age**	<16	127	28.2 (24.1–32.6)	2.37 (1.98–2.76)
	≥16	323	71.8 (67.4–75.9)	2.48 (2.24–2.72)
	Missing	313		322
**Number of oral sexes received**	>3	219	48.7 (44.0–53.4)	2.52 (2.24–2.81)
	≤3	231	51.3 (46.6–56.0)	2.37 (2.08–2.66)
	Missing	313		322
**Had sexual intercourse**	Yes	644	95.3 (93.4–96.7)	2.32 (2.15–2.49)
	No	32	4.7 (3.3–6.6)	2.22 (1.60–2.84)
	Missing	87		102
**Had sexual intercourse age**	<16	271	42.5 (38.7–46.5)	2.30 (2.04–2.56)
	≥16	266	41.8 (37.9–45.7)	2.33 (2.11–2.55)
	Missing	126		140
**Number of sexual intercourses with**	≥4	421	66.3 (62.5–70.0)	2.45 (2.24–2.67)
	<4	214	33.7 (30.0–37.5)	2.09 (1.82–2.36)
	Missing	128		143
**Current relationship status**	Stable and long	356	51.8 (48.0–55.6)	2.35 (2.13–2.57)
	Short-term	31	4.5 (3.1–6.3)	2.23 (1.44–3.02)
	Single	300	43.7 (39.9–47.5)	2.25 (2.01–2.50)
	Missing	76		91
**HPV knowledge and related behavior**				
**Ever been diagnosed with HPV**	Yes	18	2.4 (1.4–3.7)	**4.33 (3.16–5.50)**
	No	612	81.2 (78.2–83.9)	**2.23 (2.07–2.39)**
	Unknown	124	16.4 (13.9–19.3)	**2.33 (1.94–2.73)**
	Missing	9		25
**Had heard of HPV**	Yes	421	56.9 (53.2–60.5)	**2.90 (2.70–3.11)**
	No	235	31.8 (28.4–35.2)	**1.58 (1.34–1.81)**
	Not sure	84	11.4 (9.2–13.9)	1.37 (0.99–1.75)
	Missing	23		23
**Self-rated HPV knowledge**	Never heard/very poor	288	37.7 (34.8–41.9)	**1.30 (1.10–1.49)**
	Poor/fair	370	48.5 (45.6–52.8)	**2.78 (2.57–2.99)**
	Good/very good	84	11.0 (9.0–13.6)	**3.71 (3.29–4.14)**
	Missing	21		21

Note: Bold were donated as statistically significant differences.

**Table 2 ijerph-21-00307-t002:** Sample demographic characteristics, HPV knowledge and related behaviors, and associations with the prevalence of oral HPV 16/18 infection and HPV vaccination among Indigenous Australian adults.

		Oral HPV 16/18 Infection	Had HPV Vaccine
		% (95% CI)	
**Sociodemographic Characteristics**			
**Age group**	≥37	5.21 (3.08–7.34)	**6.61 (3.99–9.23)**
	<37	4.11 (1.99–6.22)	**52.71 (46.79–58.62)**
**Gender**	Male	5.76 (2.81–8.71)	**13.16 (8.31–18.01)**
	Female	4.23 (2.49–5.97)	**33.10 (28.66–37.54)**
**Location**	Non-metro	**2.59 (1.14–4.04)**	28.70 (23.80–33.60)
	Metro	**8.05 (4.95–11.16)**	25.70 (19.80–31.60)
**HPV Knowledge and Related Behavior**			
**Ever been diagnosed with HPV**	Yes	5.56 (−6.1717.28)	46.67 (18.0775.26)
	No	4.74 (3.056.43)	28.07 (23.9332.21)
	Unknown	4.84 (1.018.67)	26.56 (15.4437.68)
**Had heard of HPV**	Yes	5.70 (3.487.92)	**35.51 (30.4940.54)**
	No	3.83 (1.366.30)	**14.29 (9.3419.23)**
	Not sure	3.57 (−0.487.62)	20.63 (10.3630.91)
**Self-rated HPV knowledge**	Never Heard/very poor	5.21 (2.637.79)	**17.54 (12.5722.52)**
	Poor/fair	4.86 (2.667.07)	**33.44 (28.1738.71)**
	Good/very good	3.57 (−0.487.62)	31.94 (20.9142.98)

Note: Bold were donated as statistically significant differences.

**Table 3 ijerph-21-00307-t003:** Multivariable models for the mean HPV-KT 7 scores among Indigenous Australian adults (MR ^a^, 95% CI ^b^).

		Model 1 (Unadjusted)	Model 2	Model 3	Model 4	Model 5	Model 6
		MR (95% CI)					
Had HPV vaccine	No	0.66 (0.56–0.79)				0.79 (0.70–0.88)	0.85 (0.69–1.03)
	Yes	ref				ref	ref
Oral HPV 16/18 infection	Yes	1.04 (0.73–1.49)				1.05 (0.83–1.32)	0.88 (0.58–1.33)
	No	ref				ref	ref
Sociodemographic Characteristics							
Age group	≥37	ref	ref				ref
	<37	1.16 (1.05–1.27)	1.23 (1.16–1.37)				0.98 (0.81–1.19)
Gender	Male	ref	ref				ref
	Female	1.20 (1.08–1.34)	1.24 (1.11–1.40)				1.29 (1.01–1.54)
Location	Non-metro	ref	ref				ref
	Metro	1.05 (0.95–1.16)	1.07 (0.96–1.18)				1.13 (0.96–1.34)
Education level	High school or less	ref	ref				ref
	Trade or over	1.60 (1.33–1.61)	1.38 (1.24–1.53)				1.15 (0.96–1.37)
Income	Centrelink	ref	ref				ref
	Job	1.41 (1.28–1.57)	1.30 (1.11–1.51)				1.09 (0.85–1.40)
Health care card holder	Yes	ref	ref				ref
	No	1.29 (1.16–1.44)	1.01 (0.86–1.18)				1.00 (0.78–1.28)
Car ownership	No	ref	ref				ref
	Yes	1.31 (1.03–1.25)	1.00 (0.89–1.13)				0.97 (0.80–1.19)
Health and Related Behaviors							
Smoking status	Current smoker	0.86 (0.77–0.97)		0.81 (0.72–0.92)			0.87 (0.69–1.08)
	Ex-smoker	1.12 (0.97–1.31)		1.08 (0.91–1.27)			0.90 (0.71–1.15)
	Never smoker	ref		ref			ref
Consumes alcohol	Daily	0.94 (0.72–1.23)		0.91 (0.67–1.23)			1.03 (0.67–1.58)
	Weekly	1.00 (0.88–1.13)		0.96 (0.84–1.11)			0.83 (0.67–1.04)
	Monthly	0.98 (0.88–1.10)		0.97 (0.86–1.10)			0.96 (0.79–1.17)
	Never	ref		ref			ref
Non-tobacco Smoking	Currently	0.96 (0.82–1.12)		1.00 (0.84–1.18)			1.03 (0.78–1.36)
	Used	1.13 (1.00–1.27)		1.03 (0.90–1.18)			1.03 (0.85–1.25)
	Never	ref		ref			ref
Recreational drug use	Currently	1.14 (1.00–1.30)		1.29 (1.11–1.50)			1.25 (1.02–1.62)
	Used	1.21 (1.09–1.35)		1.29 (1.13–1.46)			1.23 (1.01–1.51)
	Never	ref		ref			ref
Ever had tonsils out	Yes	1.45 (1.12–1.88)		1.44 (1.05–1.97)			0.99 (0.81–1.22)
	No	1.30 (1.03–1.64)		1.29 (0.96–1.74)			1.20 (0.64–2.25)
	Unknown	ref		ref			ref
Self-rated general health	Excellent/very good/good	1.26 (1.12–1.42)		1.30 (1.13–1.50)			1.39 (1.11–1.75)
	Fair/poor	ref		ref			ref
Self-rated oral health	Excellent/very good/good	0.99 (0.90–1.10)		0.89 (0.79–1.00)			0.91 (0.76–1.07)
	Fair/poor	ref		ref			ref
Sexual Health Behaviors							
Had passionately kissed	≥4	1.19 (1.07–1.33)			1.29 (0.98–1.46)		0.94 (0.71–1.24)
	<4	ref			ref		ref
Ever given oral sex	Yes	1.23 (1.10–1.37)			–		–
	No	ref			–		–
Given oral sex age	<16	1.06 (0.92–1.22)			1.23 (0.98–1.55)		1.03(0.78–1.35)
	≥16	ref			ref		ref
Number of oral sexes given	>3	1.21 (1.00–1.26)			1.17 (0.95–1.43)		0.95(0.72–1.24)
	≤3	ref			ref		ref
Ever received oral sex	Yes	1.24 (1.11–1.40)			–		–
	No	ref			–		–
Received oral sex age	<16	1.05 (0.92–1.20)			0.82 (0.66–1.03)		0.94 (0.70–1.26)
	≥16	ref			ref		ref
Number of oral sexes received	>3	1.06 (0.94–1.20)			0.98 (0.80–1.21)		1.26 (0.96–1.65)
	≤3	ref			ref		ref
Had sexual intercourse	Yes	1.05 (0.82–1.33)			–		–
	No	ref			–		–
Had sexual intercourse age	<16	1.02 (0.91–1.13)			0.97 (0.82–1.15)		1.07 (0.86–1.33)
	≥16	ref			ref		ref
Number of sexual intercourses with	≥4	1.17 (1.05–1.31)			0.81 (0.66–0.99)		0.92 (0.70–1.22)
	<4	ref			ref		ref
Current relationship status	Stable and long	1.05 (0.82–1.35)			1.09 (0.95–1.25)		0.95 (0.79–1.13)
	Short-term	1.01 (0.78–1.30)			1.05 (0.77–1.43)		1.09 (0.75–1.61)
	Single	ref			ref		ref
HPV Knowledge and Related Behaviors							
Ever had HPV infection	Yes	1.86 (1.45–2.39)				1.26 (0.94–1.65)	1.09 (0.75–1.59)
	No	0.96 (0.84–1.09)				0.88 (0.76–1.04)	0.92 (0.72–1.18)
	Unsure	ref				ref	ref
Had heard of HPV	Yes	2.12 (1.75–2.57)				1.43 (1.16–1.76)	1.74 (1.19–2.55)
	No	1.15 (0.94–1.42)				1.07 (0.86–1.34)	1.28 (0.83–1.97)
	Unsure	ref				ref	ref
Self-rated HPV knowledge	Poor/fair	2.14 (1.83–2.51)				1.71 (1.48–1.96)	1.89 (1.50–2.39)
	Good/very good	2.86 (2.29–3.57)				2.23 (1.87–2.65)	2.40 (1.75–3.30)
	Never heard/very poor	ref				ref	ref

Notes: Model 1: crude model; Model 2: sociodemographic characteristics; Model 3: health and related behaviors, Model 4: sexual health, Model 5: HPV vaccine, oral HPV 16 and 18 infection, HPV knowledge and related behaviors, Model 6: full model. ^a^: MR: Mean ratio; ^b^: 95% CI: 95% confidence interval. Bold were donated as statistically significant differences.

## Data Availability

The datasets generated and/or analyzed during the current study are not publicly available due to privacy issues of the participants. Data are available from the corresponding author on reasonable request.

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
