# Peer review of "Knowledge and Awareness of HPV, the HPV Vaccine and Cancer-Related HPV Types among Indigenous Australians"

_ijerph, 2024, doi:10.3390/ijerph21030307_

Round 1

Reviewer 1 Report

Comments and Suggestions for Authors

The authors analyzed the knowledge and awareness of HPV, HPV vaccine, and HPV-related OPC among indigenous Australians from February 2018 to January 2019. Overall, the manuscript is well written and motivated. However, there are some methodological issues that could strongly jeopardise the validity of the results.

In the methods section, it is stated that this is a longitudinal study and questions about HPV vaccination status and HPV seroprevalence were assessed both at baseline and after 12 months of follow up. However, it is not clear which of these measurements were used for the analyses. If this is not a before-after study, it should be clearly described in the methods, including the study design and data used.

The description of the questionnaire and its analysis is quite poor. Please describe the properties of the HPV knowledge score and how the questionnaires were analized, including wheter and how compleatness and HPV awareness ascertainment were performed.

Minor issues:

Please delete bullet points from the results.

Use ‘participants characteristics’ rahther than ‘participants proportions’.

Delete ‘HPV-related ORL cancers’ from the title since the questionnaire does not asses knowledge about it.

Provide a reference about ARCPOH HPV study (website or if there are already pubblished data).

Results – results should not be a repetition of the information presnt in the tables, espessially the table about habits, rather try to summarize logically the most important results. The paragraph 3.5 is too long.

Please add the references to the tables in the relevant sections of the results.

Table 3. - Model 6 not 16.

Line 279-301 – this sentence needs to be rephrased.

Line 302 – what is ‘hihger score 7’’? please harmonise the score name throughout the manuscript, currently it is HPV-KT, score 7, HPV-KT 7 and 7-item HPV-KT. It is better to use the full name of the score or just the HPV knowledge score, it is not necessary to repeat how many items the questionnare has.

The first two paragraphs of the discussion are the pure repetition of the results. The first part of the discussion should be dedicated to a summary of the main results in the way that reader can already have impression about meaning, direction and importance of these results.

Line 323-324- not clear

Please harmonise past and present tense throught the results and discussion, past tense is prefered in these cases.

Line 359-360 - this sentence is just a hypothesis that has nothing to do with results of this study. Either provide a relevant reference or delete this paragraph.

Line 365 – i would say that sample size is quite low given that it covers only 5% of the Indigeneous Australian population (data from the website https://health.adelaide.edu.au/arcpoh/our-research/our-current-projects/indigenous-australian-hpv-cohort-study).

Line 367 – what does it mean ‘non-representative findings or biased estimated’? you intend selection bias and low generalisability of results?

Line 380-381 – this conclusion is contradictory. In the litterature, awareness and knowledge are two distinct entities since one person can be aware of the existance of the HPV infection (having heard about it) but may not know anything about its occurence, transmission, or health consequesnces. Therefore, participants cannot be unawere of the virus but have a ‘correct’ knowledge. Additionally, some researchers prefer to analyze the knowledge score only in the subgroup of participants who are aware of the virus. Othervise a high knowledge score among anawere individuals may indicate a low understanding of the questions and a low validity of the questionnaire.

Line 382-388 - conclusion also should not be a repettition of the results but they should rather represent an implication for the practice or restate the research problem.

Author Response

  1. The authors analyzed the knowledge and awareness of HPV, HPV vaccine, and HPV-related OPC among indigenous Australians from February 2018 to January 2019. Overall, the manuscript is well written and motivated. However, there are some methodological issues that could strongly jeopardise the validity of the results.

Thanks 

  1. In the methods section, it is stated that this is a longitudinal study and questions about HPV vaccination status and HPV seroprevalence were assessed both at baseline and after 12 months of follow up. However, it is not clear which of these measurements were used for the analyses. If this is not a before-after study, it should be clearly described in the methods, including the study design and data used.

The ‘HPV vaccination’ paragraph has been re-written (Line 149-152, pages 3-4)   

  1. The description of the questionnaire and its analysis is quite poor. Please describe the properties of the HPV knowledge score and how the questionnaires were analized, including wheter and how compleatness and HPV awareness ascertainment were performed.

We have re-written this paragraph (Line 126-143, page 3)  

  1. Minor issues:
    • Please delete bullet points from the results.

Done.

  • Use ‘participants characteristics’ rahther than ‘participants proportions’.

We have changed. 

  • Delete ‘HPV-related ORL cancers’ from the title since the questionnaire does not asses knowledge about it.

The title has been changed to ‘Knowledge and Awareness of HPV, the HPV Vaccine and cancer- related HPV types among Indigenous Australian? As we have assessed and reported HPV16, 18 associated with HPV-KT scores.

  • Provide a reference about ARCPOH HPV study (website or if there are already pubblished data).

A reference has been added (Line 112, page 3) 

  • Results – results should not be a repetition of the information presnt in the tables, espessially the table about habits, rather try to summarize logically the most important results. The paragraph 3.5 is too long.

We have updated and re-written most paragraphs in Results section.

  • Please add the references to the tables in the relevant sections of the results.

We have added. 

  • Table 3. - Model 6 not 16.

We have changed. 

  • Line 279-301 – this sentence needs to be rephrased.

We have changed.

  • Line 302 – what is ‘hihger score 7’’? please harmonise the score name throughout the manuscript, currently it is HPV-KT, score 7, HPV-KT 7 and 7-item HPV-KT. It is better to use the full name of the score or just the HPV knowledge score, it is not necessary to repeat how many items the questionnare has.

We have revised it as ‘7-item HPV-KT’ and added a statement that 7-item HPV-KT will be referred to as HPV-KT and have changed all references to this (line 136, page 4).

  • The first two paragraphs of the discussion are the pure repetition of the results. The first part of the discussion should be dedicated to a summary of the main results in the way that reader can already have impression about meaning, direction and importance

We have re-written the paragraphs. 

  • Line 323-324- not clear

We have re-written the sentences. 

  • Please harmonise past and present tense throught the results and discussion, past tense is prefered in these cases.

Thanks, all changed to past tense.

  • Line 359-360 - this sentence is just a hypothesis that has nothing to do with results of this study. Either provide a relevant reference or delete this paragraph.

We have deleted the sentence.

  • Line 365 – i would say that sample size is quite low given that it covers only 5% of the Indigeneous Australian population (data from the website https://health.adelaide.edu.au/arcpoh/our-research/our-current-projects/indigenous-australian-hpv-cohort-study).

As Aboriginal and Torres Strait Islanders comprise 3.2% of the total Australian population. Census data indicates approximately 22,000 Aboriginal adults reside in South Australia. So, our large convenience sample (n=1011) represented Indigenous Australian adults. 

  • Line 367 – what does it mean ‘non-representative findings or biased estimated’? you intend selection bias and low generalisability of results?

We have deleted this sentence. 

  • Line 380-381 – this conclusion is contradictory. In the litterature, awareness and knowledge are two distinct entities since one person can be aware of the existance of the HPV infection (having heard about it) but may not know anything about its occurence, transmission, or health consequesnces. Therefore, participants cannot be unawere of the virus but have a ‘correct’ knowledge. Additionally, some researchers prefer to analyze the knowledge score only in the subgroup of participants who are aware of the virus. Othervise a high knowledge score among anawere individuals may indicate a low understanding of the questions and a low validity of the questionnaire.

We have re-written the sentence. 

  • Line 382-388 - conclusion also should not be a repettition of the results but they should rather represent an implication for the practice or restate the research problem.

We have deleted the values. 

Reviewer 2 Report

Comments and Suggestions for Authors

1. No ethics permission for data collection.

2. Many demographic data are provided, some irrelevant or irrelevant in the study context.

3. The study was conducted in a very geographically and culturally compact group of individuals. The generalizability of these findings to a larger cohort is unclear.

4. It is nothing new that poor information dissemination promotes reluctance to vaccinate.

Comments on the Quality of English Language

1. No ethics permission for data collection.

2. Many demographic data are provided, some irrelevant or irrelevant in the study context.

3. The study was conducted in a very geographically and culturally compact group of individuals. The generalizability of these findings to a larger cohort is unclear.

4. It is nothing new that poor information dissemination promotes reluctance to vaccinate.

Author Response

  1. No ethics permission for data collection.

We had ethics permission for data collection (see ‘Institutional Review Board Statement’, Line 338-340, page 13) 

  1. Many demographic data are provided, some irrelevant or irrelevant in the study context.

An individual’s level of education and income affect their perception of HPV and HPV vaccine. While some demographic variables, such as ‘car ownership’ and  ‘HCC’, were directly or indirectly related to education level and household income , meaning they are  valuable variables to keep.

  1. The study was conducted in a very geographically and culturally compact group of individuals. The generalizability of these findings to a larger cohort is unclear.

The study (HPV-KT) was the first to be generated in a large Indigenous population (n=1011). We believe our developed 7-item HPV-KT and findings will have potential translation opportunities to other high-risk Indigenous groups at an international level. 

  1. It is nothing new that poor information dissemination promotes reluctance to vaccinate.
  • We agree and have added a sentence regarding this (Line 88-89, page 2)
  • We aimed to quantifying knowledge levels and determining factors associated with this.

Round 2

Reviewer 2 Report

Comments and Suggestions for Authors

no comments

Comments on the Quality of English Language

no comments